# Regenerative Effect of Mesenchymal Stem Cell on Cartilage Damage in a Porcine Model

**DOI:** 10.3390/biomedicines11071810

**Published:** 2023-06-24

**Authors:** Sheng-Chuan Lin, Sankar Panthi, Yu-Her Hsuuw, Shih-Hsien Chen, Ming-Ju Huang, Martin Sieber, Yan-Der Hsuuw

**Affiliations:** 1Department of Tropical Agriculture and International Cooperation, National Pingtung University of Science and Technology, No. 1, Xuefu Rd, Neipu Township, Pingtung 91201, Taiwan; vetseantw@yahoo.com.tw (S.-C.L.); b10622074@g4e.npust.edu.tw (S.P.); y9294@ymail.com (Y.-H.H.); 2Deng Chuan Animal Hospital, Kaohsiung 81361, Taiwan; 3Mercy Animal Hospital, Kaohsiung 81361, Taiwan; jillhuang31@gmail.com; 4BIONET Corp., Taipei 11497, Taiwan; martinsieber@bionetcorp.com; 5Department of Biological Science and Technology, National Pingtung University of Science and Technology, Pingtung 91201, Taiwan

**Keywords:** cartilage repair, stem cell, porcine, osteoarthritis (OA), xenogeneic

## Abstract

Osteoarthritis (OA) is a major public and animal health challenge with significant economic consequences. Cartilage degradation plays a critical role in the initiation and progression of degenerative joint diseases, such as OA. Mesenchymal stem cells (MSCs) have become increasingly popular in the field of cartilage regeneration due to their promising results. The objective of this preclinical study was to evaluate the regenerative effects of mesenchymal stem cells (MSCs) in the repair of knee cartilage defects using a porcine model. Seven healthy LYD breed white pigs, aged 9–10 weeks and weighing approximately 20 ± 3 kg, were used in the experimental protocol. Full-thickness defects measuring 8 mm in diameter and 5 mm in depth were induced in the lateral femoral condyle of the posterior limbs in both knee joints using a sterile puncture technique while the knee was maximally flexed. Following a 1-week induction phase, the pig treatment groups received a 0.3 million/kg MSC transplant into the damaged knee region, while the placebo group received a control solution as a treatment. Magnetic resonance imaging (MRI), computerized tomography (CT), visual macroscopic examination, histological analysis, and cytokine concentration analysis were used to assess cartilage regeneration. The findings revealed that human adipose-derived mesenchymal stem cells (hADSCs) were more effective in repairing cartilage than pig umbilical cord-derived mesenchymal stem cells (pUCMSCs). These results suggest that MSC-based treatments hold promise as a treatment option for cartilage repair, which aid in the treatment of OA. However, further studies with larger sample sizes and longer follow-up periods are required to fully demonstrate the safety and efficacy of these therapies in both animals and humans.

## 1. Introduction

Knee osteoarthritis (OA) is a degenerative joint disease that affects both humans and animals, resulting in significant pain and disability with economic loss [1,2,3]. The progressive degradation of articular cartilage is a significant process for the progression of OA. The loss of cartilage can result from a range of factors, including age, injury, and obesity, which can all contribute to the degenerative process. The reduction in cartilage thickness and quality leads to increased friction and pressure on the underlying bone, leading to further damage and a vicious cycle of cartilage loss and joint deterioration. Knee osteoarthritis can be categorized into primary or secondary forms based on its etiology [4,5]. The joints that are mostly affected by OA are the knee, hand, feet, and spinal joints. Osteoarthritis affects 10 to 15% of all adults over the age of 60 years, and the proportion of women compared to men is higher [6,7,8]. Recently, it estimated that osteoarthritis (OA) affects a substantial global population, with an estimated incidence of 300 million individuals specifically in the hip and knee joints [9]. The hip and knee joints represent the predominant sites of osteoarthritis burden within society [10]. Additionally, osteoarthritis is the most common form of arthritis in humans and in dogs. Dogs are more prone to developing arthritis among all domestic and pet animal species due to factors such as excessive running, injury, and/or a genetic predisposition [11,12]. Furthermore, osteoarthritis is a rapidly emerging modern disease [13].

The pathogenesis of knee osteoarthritis is significantly influenced by cartilage degradation, which plays a crucial role in disease initiation and progression. Consequently, there has been considerable research focused on investigating strategies for cartilage repair and regeneration in the context of knee OA [14,15]. The use of mesenchymal stem cells (MSCs) for cartilage repair in osteoarthritic pig models has been previously studied. Wu et al. [16] combined human umbilical cord-derived MSCs (HUCMSCs) with HA hydrogel and discovered substantial improvements in hyaline cartilage regeneration when compared to controls [16]. A study by Seto et al. [17] investigated the direct transplantation of mesenchymal stem cells into the knee joints of Hartley strain guinea pigs with spontaneous osteoarthritis. The findings from this study strongly indicate that the intra-articular injection of a hyaluronic acid (HA)–MSC mixture holds significant potential as a beneficial treatment approach for osteoarthritis [17]. Sheu et al. [18] also investigated the efficiency of cartilage transplantation with platelet-rich fibrin (PRF) augmentation in a porcine model and discovered that PRF had a beneficial effect on cartilage regeneration, with autologous chondrocytes improving the efficacy even further [18]. These studies demonstrated the potential of MSC on cartilage regeneration. Potential research gaps in these studies include the need for larger sample sizes and longer follow-up periods to fully evaluate and observe the safety and efficacy of MSC-based treatments for cartilage regeneration in OA affected pig models. Additionally, there is a need for studies to directly compare the effectiveness of different MSC sources and delivery methods in promoting cartilage regeneration. Furthermore, the use of xenogeneic mesenchymal stem cell studies in pig models, particularly in connection to osteoarthritis models, is currently limited [19].

Despite significant research efforts, knee osteoarthritis remains a major public health challenge with no known cure. Current treatment options, such as pain management and joint replacement, only offer temporary relief and do not address the underlying cartilage loss and joint degeneration [20]. Furthermore, the use of pharmaceutical therapies, such as pain relievers and oral anti-inflammatory drugs, are limited in their efficacy and can also cause significant adverse side effects and long-term complications [21,22]. Traditional methods of treating OA do not provide native cartilage [23]. However, research has found that mesenchymal stem cells have demonstrated significant potential for cartilage repair in animal studies [24]. Mesenchymal stem cells were originally isolated from bone marrow, but recent studies have identified several alternative sources, including adipose tissue, umbilical cord blood, and the placenta [25].

The current research on the effect of mesenchymal stem cells on cartilage regeneration in pigs is limited, necessitating additional studies employing large-animal models to completely investigate the safety and efficiency of MSCs in a preclinical setting [26]. The aim of this study was to investigate the efficacy of human adipose-derived MSCs and pig umbilical-derived MSCs for cartilage regeneration to treat knee defects in pigs.

The research is organized as follows to provide a comprehensive structural overview of our study: an introduction section that includes a detailed literature review and background information, a materials and methods section that outlines the reagents and equipment used in the experimental design, a results section that presents our findings using tables, graphs, and microscopy images, and a discussion section that explains the theoretical basis of our results.

## 2. Materials and Methods

The experiment titled “Regenerative effect of mesenchymal stem cell on knee cartilage damage in a porcine model” was conducted from 2022 to 2023 at the National Pingtung University of Science and Technology (NPUST), Taiwan (22°38′16.19″ N and 120°35′26.39″ E). A schematic illustration of the experimental design is presented (Figure 1), with the objective of making it more accessible to readers from diverse fields, thereby ensuring easier comprehension and understanding. The methodology for this study was as follows.

### 2.1. Preparation of MSCs

MSCs were funded by Bionet company, (Taipei City, Taiwan). They were unable to reveal the specific culture medium chosen for the study due to its unique element, which was a valuable trade secret for the company. With minor modifications to the protocol [27], human adipose-derived mesenchymal stem cells were isolated and collected after receiving informed consent. Briefly, lipoaspirates were rinsed in phosphate-buffered saline (PBS) and digested with an enzyme, such as collagenase, to break down the tissue. The stromal vascular fraction (SVF) is subsequently separated from the other cells by centrifugation. ASCs were then grown in a suitable condition from the SVF. When there were sufficient number of ASCs, they were collected for use. The method presented here with minor modification was used to isolate pig umbilical cord mesenchymal stem cells [28]. At first, the umbilical cords were disinfected with 75% ethanol, followed by a PBS wash to remove any blood contamination. Following that, the blood vessels were excised to avoid endothelial cell contamination, and the cord tissue was divided into small 0.5–1 mm^3^ fragments. These fragments were placed on 10 cm culture dishes and cultured in alpha-MEM (GIBCO, Grand Island, NY, USA) supplemented with 5% UltraGROTM (AventaCell, Atlanta, GA, USA) and antibiotics (PSA, GIBCO). The cultures were incubated at 37 degrees Celsius in a humidified atmosphere with 5% CO_2_. Every 3–4 days, new media was introduced. When the cultures reached confluence, they were washed with DPBS, harvested with 0.05% TrypLE (GIBCO, USA), and transferred to fresh 10 cm culture dishes at a plating density of 3–6 × 10^3^ cells per cm^2^ for continued growth. Finally, the MSCs were cryopreserved in culture media containing 10% DMSO in a vapor phase liquid nitrogen tank at −190 °C using a control rate freezer (Icecube, Sylab, Purkersdorf, Austria). Both hADSC and pUCMSC cells from passage three were used for the experiment. The MSCs were cultured in the BIONET Lab (No. 28, Lane 36, Xinhuyi Rd, Neihu District, Taipei City, 114065) under Good Manufacturing Practice conditions. They reported that the expanded cells were then evaluated for number, viability, purity, and identity. Before injection, MSCs were counted using a hemocytometer and loaded into a syringe.

### 2.2. Experimental Animals

The research was performed in the animal research laboratory of National Pingtung University of Science and Technology (NPUST). White LYD (the crossbred Landrace, Yorkshire, and Duroc) pigs (n = 7) body weight of 20 to 25 kg with age group 9–10 weeks were used. Pigs were purchased from a commercial farm near NPUST and placed at the NPUST positive control animal facility room. When performing this study at National Pingtung University of Science and Technology, International Animal Care and Use Committee (IACUC) protocols were followed. The International Animal Care and Use Committee (IACUC) approved the animal research and issued an IACUC permit number (NPUST-110-079). Because the ethical care of animals is of the highest concern, all pre-operative, surgical, and post-operative procedures were carried out in compliance with IACUC standards. The pigs were allowed to become acclimated to their surroundings prior to the commencement of the experiment. The pigs had unrestricted access to food and water and were reared in a controlled environment with a 12-h/12-h light/dark cycle at 23 ± 2 °C with relative humidity of 65 ± 5%. All pigs were physically evaluated to ensure their safety and well-being, and were immediately ear tagged with a specific number for identification purposes during this study (Table 1).

### 2.3. Experimental Design

In this study, we analyzed the effectiveness of the healing of cartilage using straightforward experimental designs. Twelve knees of seven pigs were randomly assigned to four groups [Table 1] (Figure 2). In the treatment group, both human adipose derived mesenchymal stem cells and pig umbilical cord derived mesenchymal stem cells were used. Each pig in the study involved a singular defect on both knees in the posterior limbs (It is important to note that no defects were created in the front limbs), which were specifically induced using a sterile puncture technique. The induced defects were carefully created in the lateral femoral condyle, with dimensions measuring 8 mm in diameter and 5 mm in depth. The knee joint was maximally flexed during the puncture procedure, as illustrated in Figure 3. For the administration of the stem cell treatment, an intra-articular injection method was employed (Figure 4). The stem cells were delivered locally and precisely to the injured area of the knee joint using this distribution method. By employing this technique, the researchers sought to increase the therapeutic efficiency of the stem cells in healing the knee abnormalities of the experimental animals.

This study was carried out in three stages, employing a consistent methodology to effectively manage workload and ensure data validity. Moreover, to address the limitations of using MRI and radiography, and to manage a large number of animals, the investigation was divided into three stages, as illustrated in Table 1. Transportation of animals to the hospital for MRI and CT examinations added to the time required for conducting the study. This approach facilitated improved planning and resource allocation, while also ensuring the welfare of the animals.

In the first stage of the experiment, two pigs were selected, with one serving as the treatment group, receiving 28 × 10^4^/kg of human adipose derived mesenchymal stem cells on both knees one week after induction surgery. The other pig acted as the placebo group and received only the medium. However, in the first round, one of the pigs received the medium solution only in the right leg due to a technical error during anesthesia [Table 1]. In the second stage, three pigs were employed, with one serving as the treatment group, receiving an average of 26 × 10^4^/kg of pig stem cells derived from umbilical cords of pigs on both knees one week after surgery. Another pig acted as the control group and received no treatment, while the remaining pig served as the placebo group, receiving 1 mL of medium only. In the third and final stage of the experiment, two pigs were selected. One pig received an average of 30 × 10^4^/kg of pig umbilical mesenchymal stem cells on both knees one week after induction surgery, while the other pig received an average of 36 × 10^4^/kg of adipose-derived mesenchymal stem cells on both knees one week after induction surgery. Through this experimental design, our objective was to evaluate the impact of human and pig stem cells on knee joint healing through cartilage regeneration, while also observing the effect of the negative control group.

### 2.4. Surgical Procedure

Zoletil^®^ 100 (virbac) along with xylazine 10%(*w*/*v*) (Health-Tech Pharmaceutical Co., Ltd., Taoyun, Taiwan) was used as anesthesia for minimally invasive surgery. The dosage was 5–10 mg/kg. Zoletil^®^ 100 was administered intravenously (IV). Isoflurane was used to maintain anesthesia in operation by the anesthesia machine. Atropine (Tai Yu Chemical& Pharmaceutical Co., Ltd., Hsinchu, Taiwan) pre anesthesia medicine was used with a dosage of 0.05 mg/kg administered subcutaneously. The antibiotic cephazolin (Sintong Taiwan Biotech Co., Taoyuan, Taiwan) was used for infection control. The dosage was 25 mg/kg twice a day, which was administered intramuscularly. Carprofen was provided to pigs after surgery as a pain relief method for knee pain. The drug’s recommended dose was 2.2 mg/kg, which was administered intramuscularly once or twice a day, depending on the severity of the wound and pain condition.

Before skin incision, betadine (Taipei, Taiwan) was applied near surgical parts. Following the skin incision, an articular cartilage defect was formed in the lateral femoral condyle of both knees. An 8 × 5 mm sterilized biopsy punch (Kai industries Co., Ltd., Seki, Japan) was used to create an incision. The wounds were sutured with 4-0 Vicryl and surgical staples (Weck Visistat ^®^ 35 W) after the procedure (See Appendix A for surgical staples) (Figure 5).

### 2.5. MRI and CT Examinations

Pre-anesthesia medication was administered to the animals; 15 min before the anesthetic dosage, atropine (0.05 mg/kg) was administered intramuscularly. Animals were induced by full dosage of mixture of Zoletil 100 (0.04 mg/kg) with xylazine (0.003/kg). Animals were taken to mercy animal hospital (Kaohsiung, Taiwan) with special care for MRI and CT examinations on every 4th and 6th weeks of stem cell implantation. Baseline was taken as 0 week before stem cell therapy (Figure 6). MRI and CT images were recorded to evaluate the articular cartilage by an experienced professor. The study used T2 mapping with frequency selective fat suppression to evaluate the condition of the cartilage. The frequency encoding direction was oriented in the anterior–posterior direction, and each measurement was repeated three times.

### 2.6. Macroscopic Images 

After eight weeks of stem cell treatments, the animals were euthanized in accordance with the protocol established by the Institutional Animal Care and Use Committee (IACUC). The animals were first administered a full dosage of a mixture of Zoletil 100 (0.04 mg/kg) and Xylazine (0.003/kg) to induce unconsciousness. Then, the knee joints were collected, pictures were taken, and evaluated using ICRS scoring [29] (Table 2). Afterwards, the knee joints were preserved in 4% paraformaldehyde for further histological examination.

### 2.7. Microscopic Evaluation

For three days, MCs from joint cartilages were preserved in 4% buffered paraformaldehyde. After two weeks of decalcification in 0.5 M ethylene diamine tetra-acetate (EDTA) solution at four degrees Celsius, the samples were desiccated, embedded in paraffin, sectioned, and stained with hematoxylin and eosin (H&E). An optical microscope (CKX41 Olympus Corporation, Tokyo, Japan) and a digital camera (EOS 80D, Canon Inc., Tokyo, Japan) were used to evaluate the histologic sections. Five independent evaluators independently analyzed the severity of articular cartilage sections and scored them using the International Cartilage Repair Society (ICRS) histology grading method [30] (Table 3). A higher score demonstrated superior quality of the repaired tissue.

### 2.8. Enzyme-Linked Immunosorbent Assay for Cytokine Concentrations

For the cytokine concentrations, blood from each group were collected; blood samples were centrifuged at 1500 g for 10 min and serum was preserved at −80 °C until use. The levels of various cytokines (interleukin-10, transforming growth factor-beta, interleukin-4, tumor necrosis factor alpha, and interleukin-1 beta) were measured using the ELISA (enzyme-linked immunosorbent assay) technique with Quantikine^®^ ELISA kits (R&D Systems, Minneapolis, MN, USA) as per the manufacturer’s instructions. The ELISA kits for all five cytokines were purchased from R&D Systems and used as per the manufacturer’s instructions. After the samples were diluted as appropriate, cytokine levels were evaluated using a four-parameter logistic (4-PL) curve fit. The results were presented as mean standard error (SEM).

### 2.9. Statistical Analysis

Descriptive statistics such as mean and standard errors (S.E.) were used to summarize the data. The ANOVA was used to determine if there were significant differences among treatment groups in terms of cytokine concentration, gross grading score, and visual histological findings score. The one-way ANOVA in IBM SPSS version 26 software was used for analysis. Multiple comparisons were performed to compare treatment means using the Duncan and Tukey post hoc tests at a statistical significance level of 5%. The graphs were created with Microsoft Excel 2022 and the most recent version of R software which is R package version 1.3-5 (Agricolae 10).

## 3. Results

### 3.1. MRI and CT Observations

This study used magnetic resonance imaging (MRI) and computerized tomography (CT) as imaging modalities to accurately assess and monitor alterations in the knee joint. Specifically, MRI and CT scans were used at the fourth and sixth weeks of the study to validate the expected localization of damage resulting from treatments. The identification of any extra cases of joint infections or injuries in addition to arthritis was also made possible by these imaging techniques. At weeks 0, 4 and 6, MRI and CT scanning were carried out (Figure 6). Week 0 was used as the baseline time for MRI and CT imaging observations in order to evaluate the defect’s initial condition prior to any intervention (Figure 7). The results obtained from magnetic resonance imaging and computerized tomography scans revealed that the group treated with human adipose derived mesenchymal stem cells demonstrated superior outcomes with respect to cartilage growth and wound healing. At week 6, it was observed that the volume of articular cartilage was lower in the defect control group, while the filling consisted of homogeneous tissue with an abnormal signal intensity of the repair tissue. Figure 8 demonstrates that the human adipose-derived mesenchymal stem cells (hADSC) group exhibits superior outcomes in terms of filling defect margins and promoting cartilage regeneration compared to the other groups, such as the pig umbilical cord-derived mesenchymal stem cells (pUCMSCs), control, and placebo group.

MRI results have been categorized according to various conditions. Sagittal views were chosen to compare each group. The T2 condition is an excellent method for demonstrating the difference between the damaged and repaired parts. The results of the MRI and CT scans showed notable improvements in the treatment group treated human adipose derived mesenchymal stem cells, especially in terms of improved cartilage formation and wound healing (Figure 8 and Figure 9). Figure 9 demonstrated that treatment group especially hADSC exhibited reduced defect margins compared to other groups. The repair stages of hADSC groups seemed better than those of pUCMSC groups. Based on the results of this study, it appears that MSCs can be used as a new way of regenerating cartilage. A viable approach to repairing articular cartilage and addressing the associated risks of OA could be achieved through this approach.

### 3.2. Gross Observations

Eight weeks after surgery, the animals were euthanized with overdose of zoletil and xyzaline and the gross appearances of the defects were taken and studied (Figure 10). The control group’s left and right knee defects had large fissures and penetration cracks. In the placebo group, smaller fissures and penetration cracks compared to control group were observed. In the treatment group (pUCMSC), the defect was covered by a thin layer of repair tissue, but the defected region was not completely covered with tissue. Surprisingly, healing was almost complete in the other treatment group (hADSC), and the reparative tissue was effectively integrated into the repair site, with flush and smooth surfaces on the restored cartilage (Figure 10). Using the International Cartilage Repair Society (ICRS) score, five independent evaluations evaluated the regeneration cartilages for coverage, neocartilage color, defect margin, and surface roughness [29]. Figure 11 demonstrates the distribution of the mean gross scores among different groups.

### 3.3. Histological Observations

Hematoxylin and eosin staining (H&E) was used to demonstrate the partial histological changes of regenerated cartilage in sample slides from all experimental groups (Figure 12). The microscopic examination of reparative tissue in the treatment group, particularly hADSC, revealed comparatively smooth restored hyaline-like cartilage with the columnar formations of chondrocytes. The findings of histological evaluations and pathological images from each experimental group indicated that cell distribution, cell population, and cartilage mineralization are better in those who were treated with human stem cells. The mean histopathology value in hADSC seems to be higher than in pUCMSC, placebo, and control groups. Several indicators, such as subchondral bone, matrix, and surface, are significantly higher in the treated group when compared to the control group (Figure 13).

The ICRS histological scoring demonstrated that hADSC had significantly higher scores (*p* < 0.05) than the control group for most parameters measured eight weeks after surgery. The placebo group had a higher mean overall score at eight weeks than the control group for surface and cartilage mineralization; however the difference was not statistically significant. Treatment groups, both hADSC and pUCMSC, had higher mean scores in some parameters, such as cell population, cell distribution, and subchondral bone. The mean score for cell population of hADSC was 1.875 ± 0.256, which was followed by pUCMSC with 1.687 ± 0.198. The mean score for the subchondral bone parameter of hADSC was 2.187 ± 0.208, which was also followed by pUCMSC with a mean score of 2.062 ± 0.192. There was a high level of agreement among the valuators. The mean histology score is shown in Figure 13.

### 3.4. Cytokine Observations

The study observed different cytokines and it revealed a significant difference in the levels of anti-inflammatory and pro-inflammatory cytokines. The results indicated that the administration of stem cells resulted in an increase in the concentration of anti-inflammatory cytokines, such as IL-4, IL-10, and TGF-beta, compared to the control group and placebo group (Figure 14, Figure 15, Figure 16, Figure 17 and Figure 18). This suggests that stem cell therapy may have a positive impact on reducing inflammation and promoting a healthy immune response. Cytokine concentrations were measured every two weeks following stem cell treatment and throughout the study (Figure 14a).

## 4. Discussion

Mesenchymal stem cells (MSCs) have garnered considerable attention as a potential source for cartilage regeneration in large animal models, including pigs [31,32,33]. In this study, we sought to evaluate the regenerative effects of two different MSC sources, namely human adipose-derived mesenchymal stem cells (hADSCs) and porcine umbilical cord derived mesenchymal stem cells (pUCMSCs), for their potential application in cartilage regeneration in pigs. The reason for selecting miniature porcine breeds as our large animal model in this study was because of their resemblance to humans in regard to joint size, loading mechanics, weight, and their inherent inability to regenerate cartilage, as well as their comparable collagen fiber arrangement, bone apposition rate, and trabecular thickness, as previously reported in previous studies [34,35]. The experimental data provided suggests that treatment with human stem cells may be a promising therapy for promoting cartilage growth and wound healing. The results obtained from MRI and CT scans demonstrated that the group treated with stem cells showed superior outcomes with respect to cartilage growth and repair. This is consistent with previous studies that have reported the potential of stem cells for regenerating cartilage tissue [36,37,38].

The concern about immune rejection was a crucial consideration in the experiment as the xenograft model was carried out. Upon conducting physical observations on the animal subjects during the study, no instances of immune rejection or tumor formation were observed in the pig’s body during the study. Gross observations at eight weeks after surgery revealed that the stem cell treatment group, particularly hADSC, had almost completely healed defects, with smooth surfaces on the restored cartilage. This contrasts with the control group, which had large fissures and penetration cracks. These findings are also consistent with previous studies that have reported the potential of stem cells for promoting tissue repair [16,23,28]. Based on the results of this study, it demonstrated that human adipose-derived mesenchymal stem cells (hADSCs) showed better results in cartilage repair compared to other groups. The histological evaluations demonstrated that cell distribution, cell population, and cartilage mineralization were better in the stem cell treatment groups compared to the control group. Specifically, the microscopic examination of reparative tissue in the hADSC group revealed smooth restored hyaline-like cartilage and more chondrocytes compared to controls (Figure 12). Better improvement in cartilage repair in the hADSC is believed to be the result of a better cell quality and cell doubling time for hADSCs compared to pUCMSCs. However, according to other findings, the basic biological characteristics of mesenchymal stem cells derived from adipose and umbilical cord tissues are comparable to one another, with both possessing significant self-renewal capacity, anti-apoptotic capacity, and multi-differentiation potential. Furthermore, investigations have discovered that the different types of cytokines generated by ASCs and UC-MSCs are comparable, although there are variations in the levels of expression of cytokines. Furthermore, several research have indicated that ASCs reacted better to various neural induction techniques than UC-MSCs, indicating that hADSCs have the potential for cartilage regeneration [39].

The process by which stem cells contribute to tissue regeneration and wound healing is underpinned by two hypotheses, namely the differentiation theory and the paracrine theory [40,41]. While both hypotheses have merit, the latter is more widely accepted and involves the concept of cell homing. This mechanism is characterized by the migration of stem cells to areas of tissue damage and subsequent secretion of cytokines that influence the behavior of adjacent cells. By promoting tissue repair and reconstruction through this paracrine mechanism, this study is also aligned with this hypothesis [42]. Specifically, this study has found that the regenerative effect of human adipose-derived mesenchymal stem cells on chondrocyte growth in pigs is associated with immune modulation, without immune rejection or tumor formation. Although this study demonstrates promising results in terms of cartilage regeneration solely using MSCs, it has been suggested by other literature by Gugliandolo et al. [43] that the bone regenerative capacity of scaffolds enriched with MSCs can be influenced and enhanced through the addition of biomolecules, such as bone morphogenetic proteins (BMPs), or the modification of biomaterial characteristics, such as pore dimensions [43]. Administration of MSCs significantly decreased the pro-inflammatory factors (TNF-alpha, or IL-1 beta), whereas the anti-inflammatory factors (IL-4 or Il-10) were significantly increased in this study, which is also supported by previous studies [15,16,34,35,36]. The dysregulation of proinflammatory and anti-inflammatory cytokines, with a predominance of proinflammatory cytokines, plays a pivotal role in driving the secretion of enzymes and other inflammatory mediators involved in the pathogenesis of osteoarthritis. This imbalance subsequently leads to detrimental morphological transformations within the joint, including cartilage degeneration, osteophyte formation, and various inflammatory alternations such as synovitis. Moreover, it seems necessary to investigate the epigenetic regulation of cytokine generation, as this could potentially lead to alternative treatment strategies for osteoarthritis [44]. This stud also investigated the impact of stem cell therapy on cytokine levels by examining three distinct anti-inflammatory cytokines (IL-4, IL-10, and TGF-beta) and two pro-inflammatory cytokines (TNF-alpha and IL-1Beta). The objective was to observe and analyze any alterations in the concentrations of these cytokines before and after the administration of stem cell therapy, thus providing insights into the therapeutic effects on cytokine profiles. The treatment group in this study was observed to have high levels of IL-10 and showed lower levels of pro-inflammatory cytokines after stem cell transplantation. This is due to IL-10 suppressing the secretion of cytokines, such as TNFα, IL-1, IL-6, IL-8, and IL-12, by dendritic cells and reducing the expression of MHC II molecules and the B7 co-stimulatory complex on their surfaces [45]. On the other hand, MSCs release many different types of cytokines, which regulate and reduce inflammatory responses, including transforming growth factor (TGF), hepatocyte growth factor (HGF), prostaglandin E2 (PGE2), soluble HLA-G5 protein, indolamine-2,3-dioxygenase (IDO), nitric oxide (NO), and interleukin-10 (IL-10) [46].

This study used a low-dose, single intra articular injection of MSCs that had no adverse effects on the experimental animals. The findings suggested a degree of safety and efficacy for a limited time as this xenograft model study is supported by the literature in terms of safety [16,47,48]. However, this study was unable to examine the results after eight weeks because the study was carried out for eight weeks after stem cell therapy. Therefore, a further study with a bigger sample number and longer duration is required to assess efficacy over a longer length of time. If the nation invests in regenerative research to develop alternative remedies for knee osteoarthritis, there is a good probability that patients’ quality of life and the country’s economy will improve.

## 5. Conclusions

The regenerative effect of human adipose MSCs on chondrocyte growth in a porcine model is associated with an immune modulation without immune rejection or tumor formation in this study. The study used a pig model to evaluate the effectiveness of mesenchymal stem cells for regenerating cartilage in the knee. According to the study’s findings, hADSCs can aid in cartilage regeneration by promoting anti-inflammatory cytokines while reducing pro-inflammatory cytokines. This study opens the door to cartilage regeneration, potentially improving treatment techniques for OA in both humans and pets. However, further studies are needed to confirm these findings and establish it as a therapeutic option.

## Figures and Tables

**Figure 1 biomedicines-11-01810-f001:**
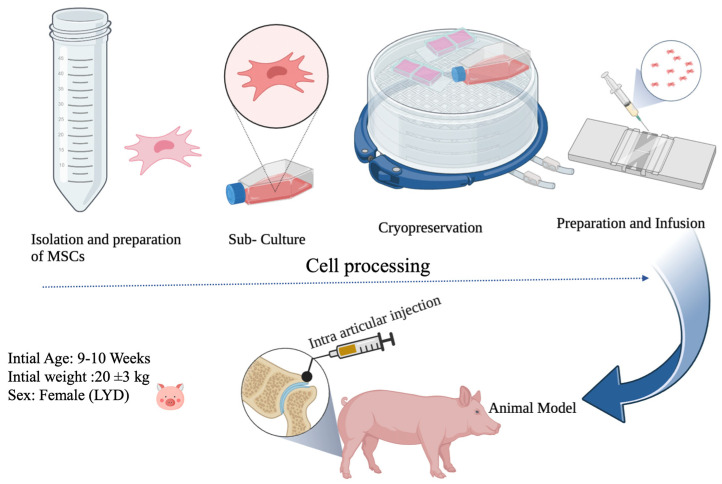
A schematic illustration of the study. (Images created in BioRender.com, accessed on 16 May 2023).

**Figure 2 biomedicines-11-01810-f002:**
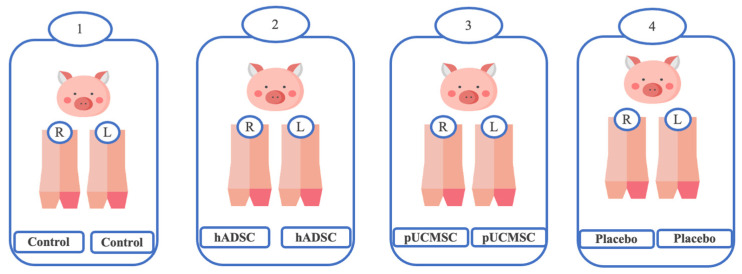
Experimental design. 1: Control group, which did not receive any intervention. 2: Treatment group receiving human adipose-derived mesenchymal stem cells. 3: Treatment group receiving pig umbilical cord-derived mesenchymal stem cells. 4: Placebo group, receiving medium only.

**Figure 3 biomedicines-11-01810-f003:**
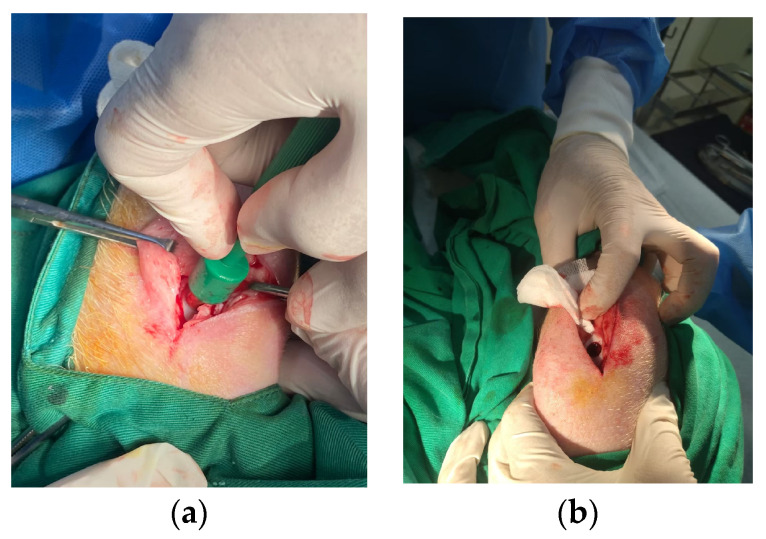
Representative pictures of osteochondral defect. (**a**) Use of sterile puncture. (**b**) Osteochondral defect created in the lateral femoral condyle of the knee.

**Figure 4 biomedicines-11-01810-f004:**
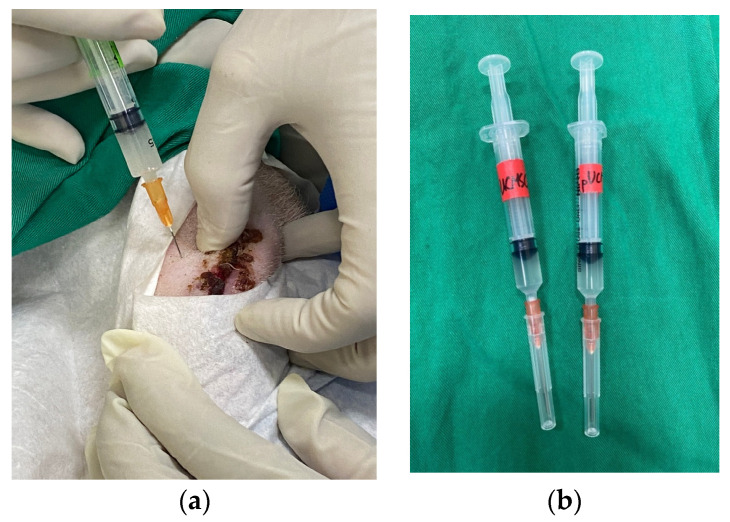
Representative images of stem cell delivery. (**a**) Stem cell implantation by intra-articular injection. (**b**) Representatives of stem cells loaded in a syringe, demonstrating the method used for cell transportation.

**Figure 5 biomedicines-11-01810-f005:**
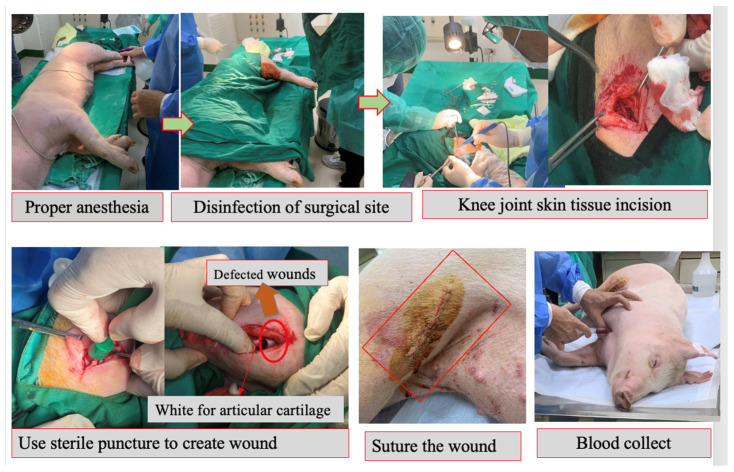
Minimally invasive surgery to create joint defects.

**Figure 6 biomedicines-11-01810-f006:**
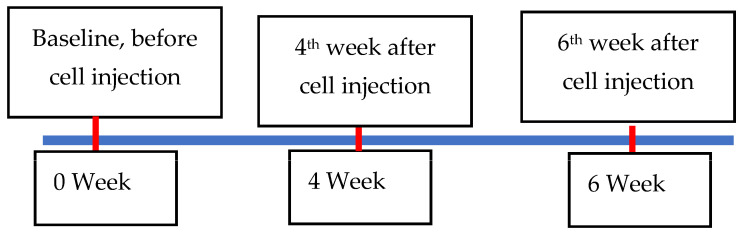
Schedule of MRI and CT examinations.

**Figure 7 biomedicines-11-01810-f007:**
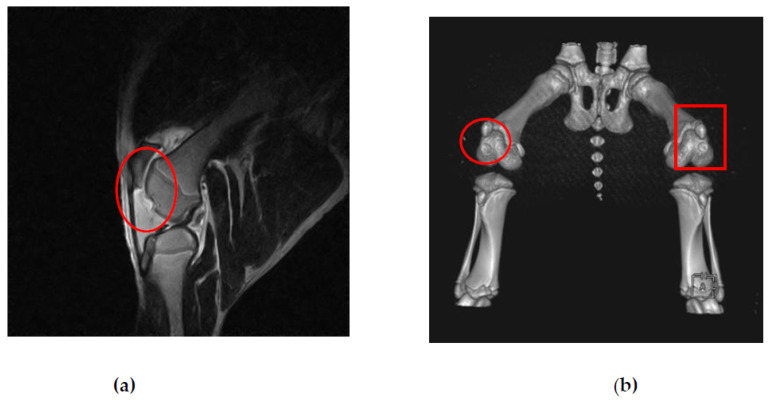
Representative MRI and CT images before cell implantation (week 0). Red circles and red box indicate the defect area. It was taken as the baseline time. MRI images taken after the induction of the knee defect but before the administration of stem cell therapy (**a**). The purpose of these images is to assess the extent of the knee defect and provide a visual representation of the initial condition before treatment. CT images taken after both knee defect before stem cell therapy (**b**). CT images provide additional information about the structural aspects of the defected knee before cell implantation.

**Figure 8 biomedicines-11-01810-f008:**
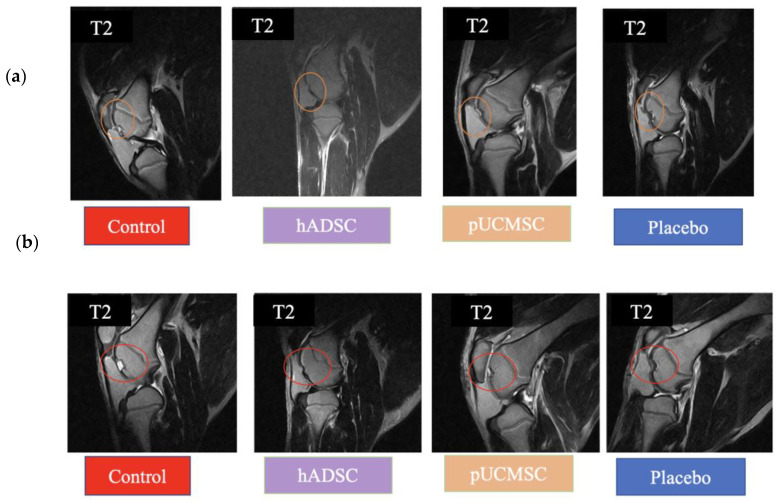
MRI examinations. (**a**) Photo images of MRI of the right knee at the 4th week of treatment using sagittal view T2 imaging. Orange circle indicates defect area. (**b**) Photo images of MRI of the right knee at the 6th week of treatment using T2 conditions. Red circle indicates defect area. On the 4th and 6th weeks of the stem cell therapy, similar results were observed from MRI exam-inations. Specifically, the group treated with human adipose-derived mesenchymal stem cells (hADSC) demonstrated superior outcomes in terms of filling defect margins and promoting carti-lage regeneration compared to the other groups.

**Figure 9 biomedicines-11-01810-f009:**
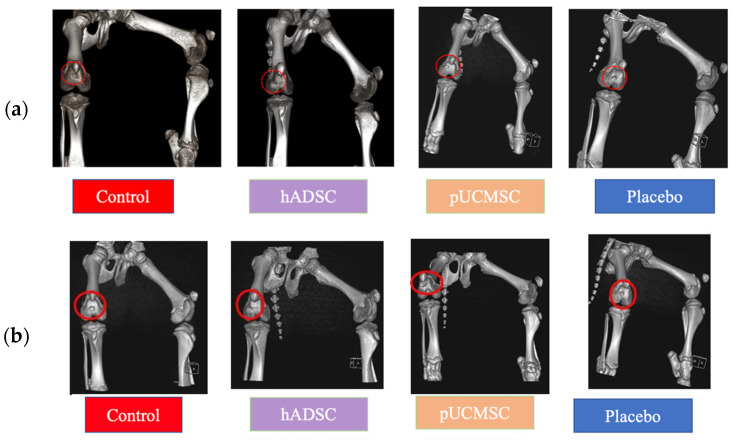
CT examinations. (**a**) Photo images of CT of the right knee at the 4th week of treatment. (**b**) Photo images of CT of the right knee at the 6th week of treatment. Red circles signify the position of the osteochondral defect filling area. During the 4th and 6th weeks of the stem cell therapy, similar results were observed through CT examinations. Specifically, the group treated with human adipose-derived mesenchymal stem cells (hADSC) exhibited reduced defect margins compared to the other groups, as evident from the CT scans.

**Figure 10 biomedicines-11-01810-f010:**
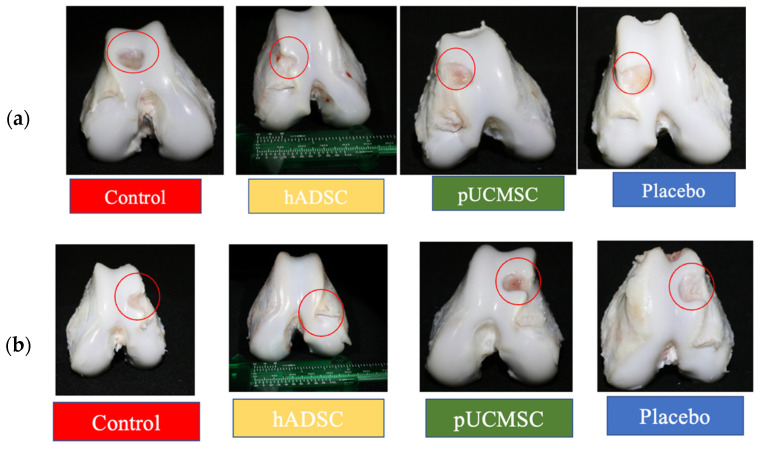
Gross evaluation of both knees from representative study groups. (**a**) Right knee joint and (**b**) left knee joint after the 8th week of treatment, respectively. Red circles signify the position of osteochondral defects. hADSC exhibited a smoother articular surface and a reduced defect margin compared to the control group.

**Figure 11 biomedicines-11-01810-f011:**
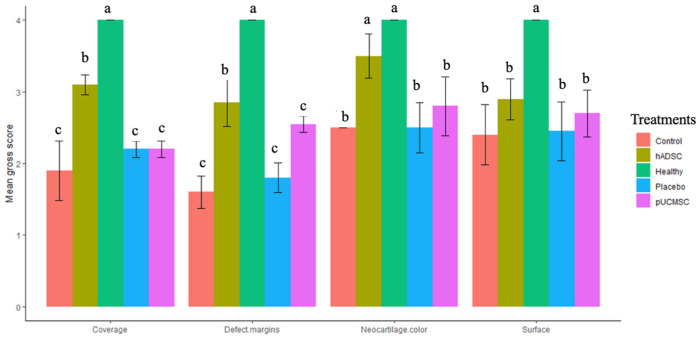
Macroscopic examination of regenerative cartilages based on overall scoring values. Error bars represent mean ± SE (n = 5). Here, n represents technical replications. The same lowercase letters are not significantly different among the experimental treatments (*p* < 0.05).

**Figure 12 biomedicines-11-01810-f012:**
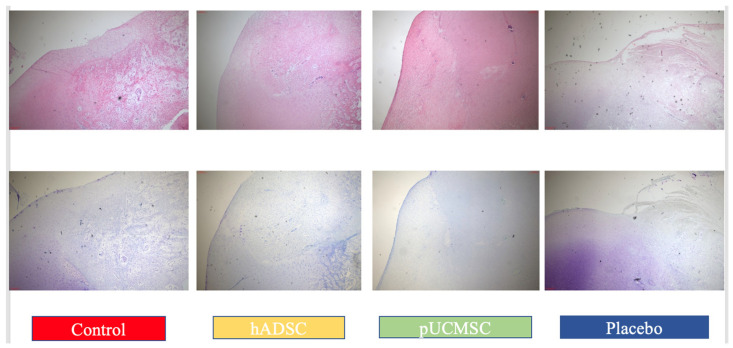
The histological appearances of regenerate cartilage in representative slides from all study groups obtained eight weeks postoperatively using hematoxylin and eosin staining, and Toluidine staining. Image magnification at 40×. The treatment group exhibited a higher number of chondrocytes cells, particularly in the human adipose-derived mesenchymal stem cell (hADSC) group, when compared to the other groups.

**Figure 13 biomedicines-11-01810-f013:**
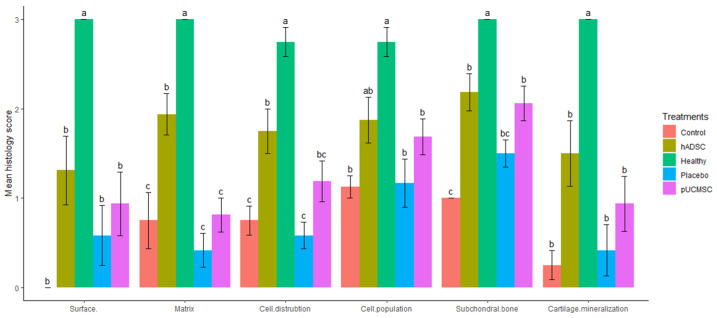
International Cartilage Repair Society (ICRS) histology scores were used to evaluate repair cartilages. The statistical error bars display the mean ± SE (n = 5). All the same lowercase letters are not significantly different between experimental treatments (*p* < 0.05).

**Figure 14 biomedicines-11-01810-f014:**
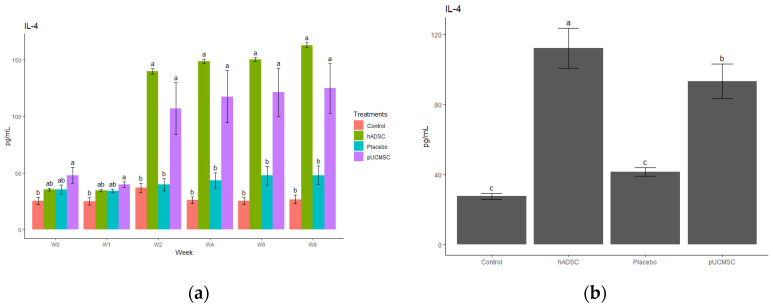
Anti-inflammatory cytokine concentration. (**a**) Determination of IL-4 cytokine concentration over time. The different time points are denoted by the x-axis, with “Wo” and “W1” signifying the week before surgery and cell implantation, respectively. The following time points, “W2”, “W4”, “W6”, and “W8”, represent two, four, six, and eight weeks after treatment in each group, respectively. The y-axis illustrates the total cytokine concentrations measured throughout the study for each group. (**b**) Total cytokines concentrations during the study in each group, where the x- axis represents different treatment groups and the y-axis represents the total cytokine concentration in picograms per milliliters (pg/mL). Cytokine concentration was measured by commercial ELISA. Error bars represent mean ± SE (n = 3). The same lowercase letters are not significantly different among the experimental treatments (*p* < 0.05).

**Figure 15 biomedicines-11-01810-f015:**
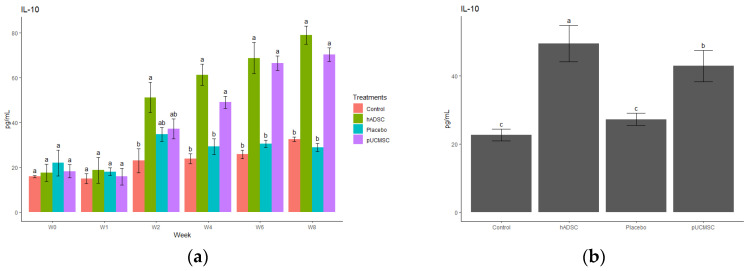
Anti-inflammatory cytokine concentration. (**a**) Determination of IL-10 cytokine concentration over time. The different time points are denoted by the x-axis, with “Wo” and “W1” signifying the week before surgery and cell implantation, respectively. The following time points, “W2”, “W4”, “W6”, and “W8”, represent two, four, six, and eight weeks after treatment in each group, respectively. The y-axis illustrates the total cytokine concentrations measured throughout the study for each group. (**b**) Total cytokine concentrations during the study in each group, where the x-axis represents different treatment groups and the y-axis represents total cytokine concentration in picograms per milliliters (pg/mL). Cytokine concentration was measured by commercial ELISA. Error bars represent mean ± SE (n = 3). The same lowercase letters are not significantly different among the experimental treatments (*p* < 0.05).

**Figure 16 biomedicines-11-01810-f016:**
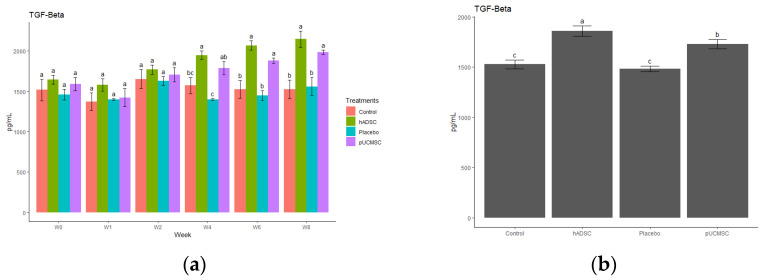
Anti-inflammatory cytokine concentration. (**a**) Determination of TGF-beta cytokine concentration over time. The different time points are denoted by the x-axis, with “Wo” and “W1” signifying the week before surgery and cell implantation, respectively. The following time points, “W2”, “W4”, “W6”, and “W8”, represent two, four, six, and eight weeks after treatment in each group, respectively. The y-axis illustrates the total cytokine concentrations measured throughout the study for each group. (**b**) Total cytokine concentrations during the study in each group, where the x- axis represents different treatment groups and the y-axis represents the total cytokine concentration in picograms per milliliters (pg/mL). Cytokines concentration was measured by commercial ELISA. Error bars represent mean ± SE (n = 3). The same lowercase letters are not significantly different among the experimental treatments (*p* < 0.05).

**Figure 17 biomedicines-11-01810-f017:**
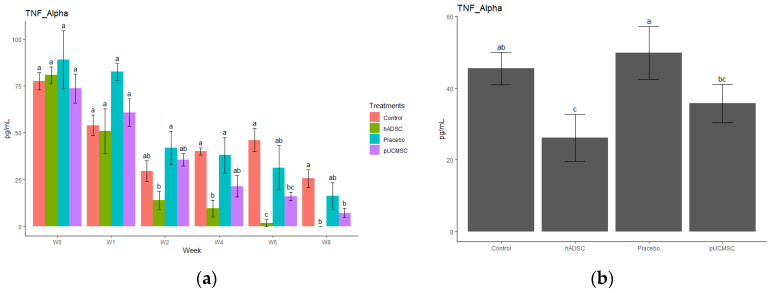
Cytokine concentration. (**a**) Determination of TNF-alpha cytokine concentration over time. The different time points are denoted by the x-axis, with “Wo” and “W1” signifying the week before surgery and cell implantation, respectively. The following time points, “W2”, “W4”, “W6”, and “W8”, represent two, four, six, and eight weeks after treatment in each group, respectively. The y-axis illustrates the total cytokine concentrations measured throughout the study for each group. (**b**) Total cytokine concentrations during the study in each group, where the x- axis represents different treatment groups and the y-axis represents the total cytokine concentration in picograms per milliliters (pg/mL). Cytokine concentration was measured by commercial ELISA. Error bars represent mean ± SE (n = 3). The same lowercase letters are not significantly different among the experimental treatments (*p* < 0.05).

**Figure 18 biomedicines-11-01810-f018:**
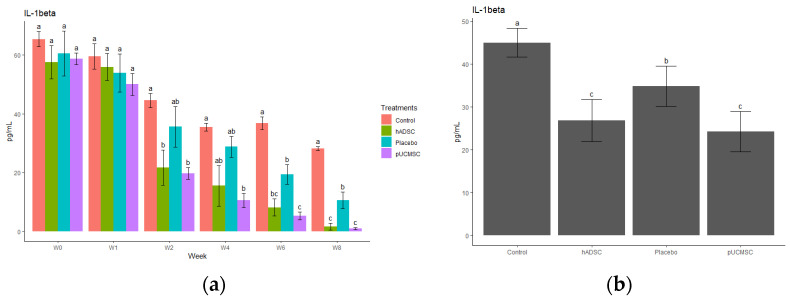
Cytokine concentration. (**a**) Determination of IL-1 beta cytokine concentration over time. The different time points are denoted by the x-axis, with “Wo” and “W1” signifying the week be-fore surgery and cell implantation, respectively. The following time points, “W2”, “W4”, “W6”, and “W8”, represent two, four, six, and eight weeks after treatment in each group, respectively. The y-axis illustrates the total cytokine concentrations measured throughout the study for each group. (**b**) Total cytokine concentrations during the study in each group, where the x-axis represents different treatment groups and the y-axis represents the total cytokine concentration in picograms per milliliters (pg/mL). Cytokine concentration was measured by commercial ELISA. Error bars represent mean ± SE (n = 3). The same lowercase letters are not significantly different among the experimental treatments (*p* < 0.05).

**Table 1 biomedicines-11-01810-t001:** Experimental design treatment groups.

Group	Ear Tag	Received
Treatment	P51, P72	hADSC ^1^
Treatment	P61, P71	pUCMSC ^1^
Control	P62	Nothing
Placebo	P52, P63	Medium

^1^ hADSC = human derived adipose mesenchymal stem cell, pUCMSC = pig umbilical mesenchymal stem cell.

**Table 2 biomedicines-11-01810-t002:** Macroscopic evaluation using the ICRS score [30].

Gross Appearance	Grade
**Coverage**	
N > 75% fill	4
50–75% fill	3
25–50% fill	2
<25% fill	1
No fill	0
**Neocartilage color**	
Normal	4
25% yellow/brown	3
50% yellow/brown	2
75% yellow/brown	1
100% yellow/brown	0
**Defect margins**	
Invisible	4
25% circumference visible	3
50% circumference visible	2
75% circumference visible	1
Entire circumference visible	0
**Surface**	
Smooth/level with normal	4
Smooth but raised.	3
Irregular 25–50%	2
Irregular 50–75%	1
Irregular > 75%	0

**Table 3 biomedicines-11-01810-t003:** Histological visual grading [30].

Features	Score
**Surface**	
Smooth/continuous	3
Discontinuities/irregularities	0
**Matrix**	
Hyaline	3
Mixture: hyaline/fibrocartilage	2
Fibrocartilage	1
Fibrous tissue	0
**Cell distribution**	
Columnar	3
Mixed: columnar/cluster	2
Cluster	1
Individual cells/disorganized	0
**Subchondral bone**	
Normal	3
Increased remodeling	2
Bone necrosis/granulation tissue	1
Detached/fracture/callus at base	0
**Cartilage mineralization (calcified cartilage)**	
Normal	3
Abnormal/inappropriate	0

## Data Availability

Data are contained within the article or Appendix A.

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
