# Peer review of "Regenerative Effect of Mesenchymal Stem Cell on Cartilage Damage in a Porcine Model"

_biomedicines, 2023, doi:10.3390/biomedicines11071810_

Round 1
Reviewer 1 Report
In this research, the authors investigated the regenerative effects of different mesenchymal stem cells on cartilage damage in a porcine model. I think this manuscript is poorly written in various aspects, including logical flow, grammar, discussion of results, figure design, and English language usage. It cannot be published in its current format. The comments are as follows:
1.Please reorganize the introduction section.
2. There are numerous inconsistencies in this manuscript, such as:
1) a) In the abstract and methods section, the defect was described as being in the central weight-bearing area. However, Figure 7 shows the defects in a different location. However, in Figure 7, the defects were close to the edge.
2) In the second stage, 26×104 /kg of pig stem cells derived from umbilical cords were applied. However, in the third stage, average of 30×104 /kg of pig umbilical mesenchymal stem cells were used.
3) Different amounts of stem cells were employed in this study, including 36×104/kg adipose-derived mesenchymal stem cells, 26×104 /kg of pig stem cells, and 30×104 /kg of pig umbilical mesenchymal stem cells. Please provide an explanation for this discrepancy.
3. It appears that the cells were applied one week after the defect surgery. In many animal model studies, cells or other agents are typically placed during surgery. Please address this difference.
4. After 8 weeks, the samples were collected for histological observations, while MRI and CT scans were taken at 4/6 weeks. Please explain why MRI and CT scans were not conducted at 8 weeks before sample collection.
5. Please double-check the reference style.
6.Additional experiments are needed to detect regenerated cartilage more comprehensively.
Extensive editing of English language required
Author Response
The manuscript details about the regenerative effect of mesenchymal stem cell on cartilage damage in porcine model. The authors would like to thank and appreciate the reviewer for committing time to providing value through some honest assessments of the manuscript's contents. The review comments and criticisms have greatly improved the work. We value your helpful comments and the time you took to evaluate our work. We sincerely regret any errors found in the manuscript, and we promise you that we have taken care of all your concerns in order to make the manuscript better overall

Reviewer 2 Report
This is the revised version of the manuscript 2376668. The authors had addressed all my comments.
Author Response
The review comments and criticisms have significantly enhanced the quality of our work. We deeply appreciate the valuable feedback and the time invested by the reviewer in evaluating our manuscript. As this reviewer did not pose any further questions, we would like to express our gratitude for the previous questions raised, which have contributed to the overall improvement of our manuscript.
Round 2
Reviewer 1 Report
The authors have addressed most of the review comments in the Excel file. I have a few minor questions.
1. There are numerous cartoons piled together in figure 1. Please organize them in a simple and clear manner.
2. The manuscript contains some comments. Has this manuscript been fully revised to the latest version?
Moderate editing of English language required
Author Response
General Comments and background
The manuscript details about the regenerative effect of mesenchymal stem cell on cartilage damage in porcine model. The authors would like to thank and appreciate the reviewer for committing time to providing value through some honest assessments of the manuscript's contents. The review comments and criticisms have greatly improved the work. We value your helpful comments and the time you took to evaluate our work. We sincerely regret any errors found in the manuscript, and we promise you that we have taken care of all your concerns in order to make the manuscript better overall.
Specific Comments (Summary of Responses to Reviewer Comments)
- One of the concerns raised by the reviewer pertained to Figure 1, where they suggested removing the cartoons that were piled together. In response to this comment, we have taken careful measures to reorganize Figure 1, which effectively illustrates the scheme of our study. (Figure 1 reorganize )
- Another question raised by the reviewers pertained to whether the manuscript had been fully revised to the latest version. In response, we have conducted another round of thorough revisions, ensuring that all necessary updates and improvements have been incorporated. Additionally, we have also focused on enhancing the English language editing of the manuscript. (Changes on whole manuscript)
